# Explainable machine learning to identify chronic lymphocytic leukemia and medication use based on gut microbiome data

Tereza Fait Kadlec,[1,2] Emma Elizabeth Ilett,[3,4] Caspar da Cunha-Bang,[1] Henrik Sengeløv,[1] Christian Brieghel,[1,2] Arda Gulay,[3] Sulman Rafiq,[5] Hanne Berg Ravn,[6,7] Chaoqun Zheng,[8] Rikke Vibeke Nielsen,[9,10] Søren Schwartz Sørensen,[11,12] Ramtin Zargari Marandi,[3] Carsten Utoft Niemann[1,2,12]

**ABSTRACT** Medication, particularly antibiotics, significantly alters gut microbiome composition, often reducing microbial diversity and affecting host health. Given that the gut microbiome may influence cancer progression, we integrated clinical, shotgun metagenomic, and medication data to assess microbiome composition across diseased and healthy cohorts, as well as the impact of medication on microbiome variation. The study cohorts included patients with chronic lymphocytic leukemia (CLL, $n = 85$), acute myeloid leukemia (AML, $n = 61$), myeloid dysplastic syndrome (MDS), and other severe hematological malignancies ($n = 104$); patients scheduled for elective cardiac surgery ($n = 89$); and kidney donors ($n = 9$), all collected as part of a consecutive microbiome sampling effort at Copenhagen University Hospital, Denmark; and healthy individuals ($N = 59$). First, our analyses revealed similarities in both diversity and composition between microbiomes of patients with CLL and patients prior to elective cardiac surgery, whereas patients with AML and MDS exhibited the least diverse and most distinct microbiomes. Second, when we quantified sources of microbiome variation, the combination of medication, disease, age, and sex accounted for 4% of variation between all cohorts and 10.4% of variation between CLL and pre-cardiac surgery patients only; the two cohorts selected for comparison due to their similar microbiomes. Notably, this left 90%–95% of the variation unexplained, emphasizing the need for better identification of the parts of the microbiome variation impacting health and disease. Third, using a machine learning approach, we validated and further refined the CLL-associated microbiome pattern from our previous studies. Overall, our data provide a foundation for further investigation into disease-specific microbial signatures and the potential interactions between medication, underlying disease, and the microbiome, with the ultimate goal to improve our understanding and clinical management of CLL.

**IMPORTANCE** This study reveals how disease and medication influence the gut microbiome in patients with chronic lymphocytic leukemia (CLL) when compared to other more severe hematological malignancies, a cohort of patients scheduled for elective cardiac surgery representing a severely diseased nonhematological cohort, and a cohort of healthy individuals. We found that patients with CLL and those scheduled for cardiac surgery had the most similar microbiome diversity and composition. Similarities across very different disease contexts suggest that disease status alone has limited impact. Consistently, across all cohorts, medication, disease, age, and sex together explained only less of microbiome variation, leaving 90%–95% unexplained. This underscores the important need for better identification of factors shaping the microbiome. In addition, we validated a previously published, machine learning-based CLL-associated microbiome signature, demonstrating the robustness of our previous findings differentiating the microbiome signature for CLL as compared to healthy individuals. The findings expand knowledge on how disease states and medical

Address correspondence to Carsten Utoft Niemann, carsten.utoft.niemann@regionh.dk.

C.U.N. received research funding and/or consultancy fees from Abbvie, Janssen, AstraZeneca, Genmab, Beigene, CSL Behring, Octapharma, Lilly, Takeda, MSD, Synamics, and Novo Nordisk Foundation but declares no financial or non-financial competing interests. C.D.C.-B. reports consulting fees from Janssen, honoraria for lectures from Octapharma, support for attending meetings from AbbVie and Octapharma, and participation on advisory boards for Janssen, BeiGene, and AstraZeneca but declares no financial or nonfinancial competing interests. The remaining authors have no conflicts of interest to declare.

treatments shape gut microbiome composition and diversity, potentially leading to new ways of managing CLL and improving patient outcomes through microbiome signatures.

**KEYWORDS** microbiome, chronic lymhocytic leukemia, medication, antibiotics, machine learning, signature

Gut microbiome research is rapidly evolving, with numerous studies identifying associations between the gut microbiome and various diseases (1–5). It has been established that the gut microbiome can be influenced by many factors, including underlying diseases, medication, diet, and exercise (6–8). Medication, particularly antibiotics, has a profound impact on microbiome composition, often resulting in reduced microbial diversity and proliferation of resistant strains (9, 10), which can have a lasting effect on host health (11, 12). Due to the importance of these different factors influencing the gut microbiome and the gut microbiome's own role in clinical outcomes, it is vital that we integrate clinical, microbial, and medication data in comprehensive studies to reveal the complexity of these interactions and dissect factors driving clinical importance. In this cross-sectional study, we combine high-quality clinical, medication, and metagenomic data and investigate different clinical gut microbiome profiles focusing on chronic lymphocytic leukemia (CLL), a complex hematological malignancy, in the context of other hematological malignancies, cardiovascular diseases, and a healthy population.

Hematological patients, including those with CLL, frequently exhibit compromised immune systems due to the nature of the disease and/or antineoplastic treatment. For CLL, it is unclear whether immune dysfunction, and thus the increased risk of infections, or the malignant CLL clone initiates the first step in tumorigenesis (13–15). Immunosuppression renders the patients particularly susceptible to infections, which remains to be one of the leading causes of death in patients with hematological malignancies (16, 17). To mitigate the increased risk of infection, antibiotics and antivirals are routinely administered as a prophylactic and/or therapeutic measure. Most patients with planned cardiac surgery included in our study received antithrombotic and lipid-modifying agents, as well as prophylactic antibiotics prior to cardiac surgery. Thus, we hypothesize that the observed microbiome profiles in our cohorts do not solely reflect underlying diseases but are influenced by medication as well.

Given that the gut microbiome has been proposed to significantly influence cancer microenvironments, immune responses, and disease progression (18–21), as well as the efficacy of targeted therapies (22, 23), our study could lead to an improved understanding and management of CLL by validating previously suggested or identifying new CLL-specific microbial signatures. Therefore, the main aim of our study is to investigate the influence of administered medication in defining these signatures.

## MATERIALS AND METHODS

### Patient cohort and healthy individuals

We included fecal samples from 409 patients (one sample per subject) enrolled in the CLL biobank and the PERSIMUNE biobank (both located at Rigshospitalet, Copenhagen, Denmark), as part of the PERSIMUNE gut microbiome research initiative (24, 25) between 2015 and 2021. We included samples from patients with CLL; patients who were scheduled to undergo allogeneic hematopoietic stem cell transplantation (aHSCT) due to acute myeloid leukemia (AML), myelodysplastic syndrome (MDS), or other hematological malignancies; patients scheduled for elective cardiac surgery; healthy living kidney transplant donors; and a publicly available cohort of Danish healthy individuals (26, 27). Samples in the CLL cohort were collected at different stages of the disease (including treatment-naïve, treated, and relapsed-refractory patients, Table S1). Only samples collected at least 5 days prior to aHSCT (due to start of the pre-transplantation conditioning regimens) were included; detailed sample collection in the pre-aHSCT groups

has been previously described by Ilett et al. (28). Samples from patients scheduled for elective cardiac surgery, included as a critically ill nonhematological contrast cohort to CLL, were collected before surgery to avoid potential microbiome changes caused by the procedure. Baseline feces samples from healthy individuals under the age of 45 years were excluded for a comparable age range of CLL patients. Cohort characteristics can be found in Table 1. The project was performed in compliance with the guidelines of the Declaration of Helsinki and approved by the Danish national ethics committee (Approval No. 1804410, Journal No. H-16047481), as well as the Danish Data Protection Agency (RH-2017-67, I-suite 05320). Written informed consent was obtained from all patients prior to sampling.

## Sample collection, sequencing, and preprocessing

As previously described (19, 28, 29), fecal samples were collected, fixated by stabilization fluid, and subsequently frozen. All samples were stored at −80°C until shipment for sequencing. All samples, including samples from healthy cohorts, underwent shotgun metagenomic sequencing on the Illumina Hi-Seq platform. After sequencing, reads underwent preprocessing and quality control steps (Supplemental Methods).

## Data sources and medication records

The information on medication usage was obtained from a review of electronic health records (EHR [30]). The medication records in this project cover medicine administered and prescribed at the hospital, as well as medicine prescribed outside the hospital (please refer to Supplemental Methods for more details). We collected medication data prescribed or administered to an individual between 30 days and ≤1 day prior to the sampling date. Only medication used by more than five individuals across assessed cohorts was considered. Data on antineoplastic treatment were collected for patients from CLL, pre-aHSCT AML, and other pre-aHSCT cohorts between 180 and ≤1 day prior to microbiome sampling.

## Taxonomical and functional profiling

Taxonomic profiling and estimation of the relative abundances at all taxonomical levels were performed using MetaPhlAn4 (31). As previously described (32), metabolic functional abundances were assessed through profiling of gut metabolic modules (GMMs). More details on how GMMs are obtained can be found in Supplemental Methods. The resulting GMM profiles reflect the predicted functional potential of the microbiome and will, therefore, be referred to as "GMM functional profiles" throughout this work.

**TABLE 1** Patient cohort characteristic[c]

| Cohort | Pre-aHSCT AML | Other pre-aHSCT | CLL | Pre-cardiac surgery | Kidney donors | Healthy |
|---|---|---|---|---|---|---|
| N per cohort | 61 | 104 | 85 | 89 | 9 | 59 |
| Median age at sampling (IQR) | 60.0 | 59.5 | 68.7 | 64.6 | 49.6 | 55.2 |
| | (51.1–67.4) | (46.8–66.0) | (62.0–73.0) | (59.7–71.1) | (45.3–58.6) | (52.6–61.3) |
| % males | 27 (44.3%) | 64 (61.5%) | 54 (63.5%) | 69 (78%) | 4 (44.4%) | 22 (37.3%) |
| % used antibiotics[a] | 61 (100%) | 65 (62.5%) | 16 (19%) | 5 (5%) | 0 | 0 |
| % received chemotherapy[b] | 61 (100%) | 66 (63%) | 4 (5%) | 0 | 0 | 0 |

[a]30 days prior to microbiome sampling.
[b]180 days prior to sampling. data on antibiotics usage and chemotherapy in the other pre-aHSCT cohort might be underrepresented due to referrals of patients between hospitals, causing insufficient data coverage. Only medication used by more than five individuals across the cohorts was considered. Eighty-five patients with CLL included 70 previously published (19, 29). The previously published cohort of 10 patients [29]), which served as the basis for our proposed CLL microbiome pattern, was excluded in microbiome pattern validation analysis (Fig. 3).
[c]Pre-aHSCT AML: patients with acute myeloid leukemia; CLL: chronic lymphocytic leukemia (Table S1); other pre-aHSCT: patients with myelodysplastic syndrome or other hematological diseases (Table S1); pre-cardiac surgery: patients before undergoing cardiac surgery; antibiotics: antibiotics or antimycotics (J01 and J02); % used: % of individuals from the cohort who had registered use of any dosage of the respective medication within the indicated time period.

## Bioinformatics and statistical analyses

Exploratory analyses for all cohorts were performed with centered-log-ratio-transformed (clr) relative bacterial abundance as the input data. As in our previous studies (19, 29), Wilcoxon rank-sum testing was used to identify significant differences between subgroups, and the Benjamini-Hochberg (BH) method was used for multiple-testing correction, with a BH-adjusted $P < 0.05$ being considered significant. PERMANOVA was used to test variations in the microbial composition among cohorts (age, gender, and sequencing batch effects). Alpha diversity measure (Shannon index) was calculated at the species level using the *vegan* R package (33).

To understand the variability and commonality in gut microbiome structures across different cohorts, enterotype identification was performed as published by Arumugam et al. (34). In short, Jensen-Shannon divergence (JSD) based on relative genus abundances was calculated between pairs of samples to create a distance matrix. Partitioning Around Medoids (PAM) clustering was performed on the distance matrix to identify clusters of samples.

The principal component analysis (PCA) was calculated using the *prcomp* R function with auto-scaled square root-transformed relative abundance of bacterial species as the input. Univariate generalized linear models (GLM) were used to screen for principal components (PCs) that might explain the binomial response (1: CLL; 0: non-CLL, significant PC: GLM $P < 0.05$). Representative centroids for each cohort were calculated by taking the mean of all significant PCs. To determine the relative similarity between cohorts, we calculated the angles between one another using cosine similarity between the centroid vectors of each cohort over 500 iterations. From the mean of the angles, we derived angular distances and the standard deviations reported in the Results section. For more details, please see the Supplemental Methods.

To assess the predictive power of different metadata and medication groups on gut microbiome variance, we used redundancy analysis with the *rda* and stepwise redundancy analysis with the *ordiR2step* functions from the *vegan* package. To identify associations between individual medication groups and microbial abundances and GMM functional profiles, multivariate regression analysis was used. The analysis involved a two-step regression analysis using the *glm2* function, as described by Nagata et al. (35). Univariate regression was used to assess the individual effect of medication, whereas multivariate regression analysis was applied to assess the combined effect of medication with all significant variables identified by univariate regression, age, and sex as explanatory variables included. No additional corrections for multiple testing were done as only variables with BH adjusted $P$-values $< 0.05$ in the univariate regression analysis were included in the multivariate regression analysis.

## Machine learning models

To perform machine learning, we used medical artificial intelligence toolbox (MAIT), a pipeline tailored for tabular data and binary classification (36). We performed fivefold cross-validation on seven models to classify CLL as an outcome. The models included logistic regression, random forest, Naïve Bayes, histogram-based gradient boosting classification tree (HistGBC), QLattice, LightGBM, and CatBoost. The ML classification included hyperparameter tuning, feature selection, and robust evaluation by multiple performance metrics (Supplemental Methods). Correlation of the features to the outcome variable was assessed by point-biserial correlation with 1,000 bootstrap (resampling with replacement) samples.

This manuscript has been prepared following the STORMS guideline, and the complete checklist can be found at: https://github.com/PERSIMUNE/PAC2025_MAIT_CLL

## RESULTS

### Patient characteristics and medication use

In this study, we included a total of 409 feces samples from 409 individuals including 85 patients with CLL (of which 70 were previously published (19, 29), cohort abbreviated as "CLL"), patients scheduled to undergo aHSCT including 61 patients with AML ("Pre-aHSCT AML") and 104 patients with MDS and other severe hematological malignancies ("Other pre-aHSCT"), 89 patients scheduled for elective cardiac surgery ("Pre-cardiac surgery"), nine healthy living kidney donors prior to donation ("Kidney donors"), and 59 healthy individuals ("Healthy" [26, 27]). Cohort characteristics are shown in Table 1 and Table S1. Medication with impact on the gut microbiome (as reported by Weersma et al. [37]) was assessed across these six cohorts and categorized at the Anatomical Therapeutic Chemical (ATC) 3rd level codes (Table 2). As anticipated, exposure to both chemotherapy and antibiotics was gradually higher among patients with CLL, other pre-aHSCT, and pre-aHSCT AML (Chi2 between-group difference: CLL vs pre-aHSCT AML, $P = 0.01$; CLL vs other pre-aHSCT, $P = 0.02$). In the CLL group, no significant difference was observed in microbiome composition with respect to the treatment status, immunoglobulin replacement therapy, or genetic biomarkers such as IGHV mutation status or TP53 mutation (Table S1).

### Microbiome diversity and analysis of covariance by PCA suggest distinct microbiomes across cohorts

Microbiome diversity was analyzed across the six cohorts, revealing that microbial diversity was significantly lower for pre-aHSCT AML patients and significantly higher for healthy individuals compared to those in all other cohorts. Remaining cohorts had in-between diversity levels and were not significantly different compared to each other (Fig. 1A; Table S2). We next attempted to deconvolute the overall contribution of microbial variation by identifying the principal components driving a generally healthy gut composition, microbiome of pre-aHSCT AML, other pre-aHSCT, and pre-cardiac surgery patients, as opposed to what might be characterized as the CLL microbiome (considered as baseline axis at 0 degrees, Fig. 1B). To assess this, we illustrated the covariance in each cohort by their centroids in PCA space (Fig. 1B). Only principal

**TABLE 2** Number of patients (not) receiving medication 30 days prior to microbiome sampling[a]

| Medication name[b] | ATC codes | CLL (n = 85) | Cardiac surgery (n = 89) | Kidney donor (n = 9) | Healthy (n = 59) |
|---|---|---|---|---|---|
| Drugs for acid-related disorders (PPIs) | A02 | 4 | 23 | 0 | 0 |
| Drugs for constipation (laxatives) | A06 | 1 | 8 | 2 | 0 |
| Drugs used in diabetes[c] | A10BA02 (metformin) | 1 | 6 | 0 | 0 |
| Antithrombotic agents[d] | B01 | 13 | 58 | 2 | 0 |
| Drugs used in cardiac therapy | C01 | 2 | 40 | 0 | 0 |
| Beta-blocking agents | C07 | 9 | 25 | 0 | 0 |
| ACE inhibitors, angiotensin II antagonists | C09 | 9 | 26 | 0 | 0 |
| Lipid-modifying agents (statins) | C10 | 7 | 52 | 0 | 0 |
| Corticosteroids | H02 | 5 | 0 | 1 | 0 |
| Psycholeptics[e] | N05 | 9 | 42 | 2 | 0 |
| Antibacterials and antimycotics | J01, J02 | 16 | 5 | 0 | 0 |
| No medication | NA[f] | 40 | 0 | 5 | 59 |

[a]One patient can receive multiple medications with the same ATC 3th group.
[b]Only medication taken by > 5 patients in 30 days prior to sampling is included.
[c]Metformin (A10BA02) was selected from the A10 medication class as the sole antidiabetic drug to be evaluated due to its well-documented impact on the gut microbiome (38), as well as to exclude the use of insulin, which has a considerably lesser-known impact on the microbiome.
[d]Acetylsalicylic acid (aspirin) ATC code is included among antithrombotic agents (B01A) in our patient records.
[e]High % of psycholeptic medication usage in pre-cardiac surgery patients is medication taken as perioperative medicine. The data on psycholeptics administration can be subject to small inconsistencies due to the missing exact microbiome sampling date in six patients. PPIs: proton pump inhibitors. Antihistamines and analgesics were excluded as they might be used in the healthy cohorts.
[f]NA, not applicable.

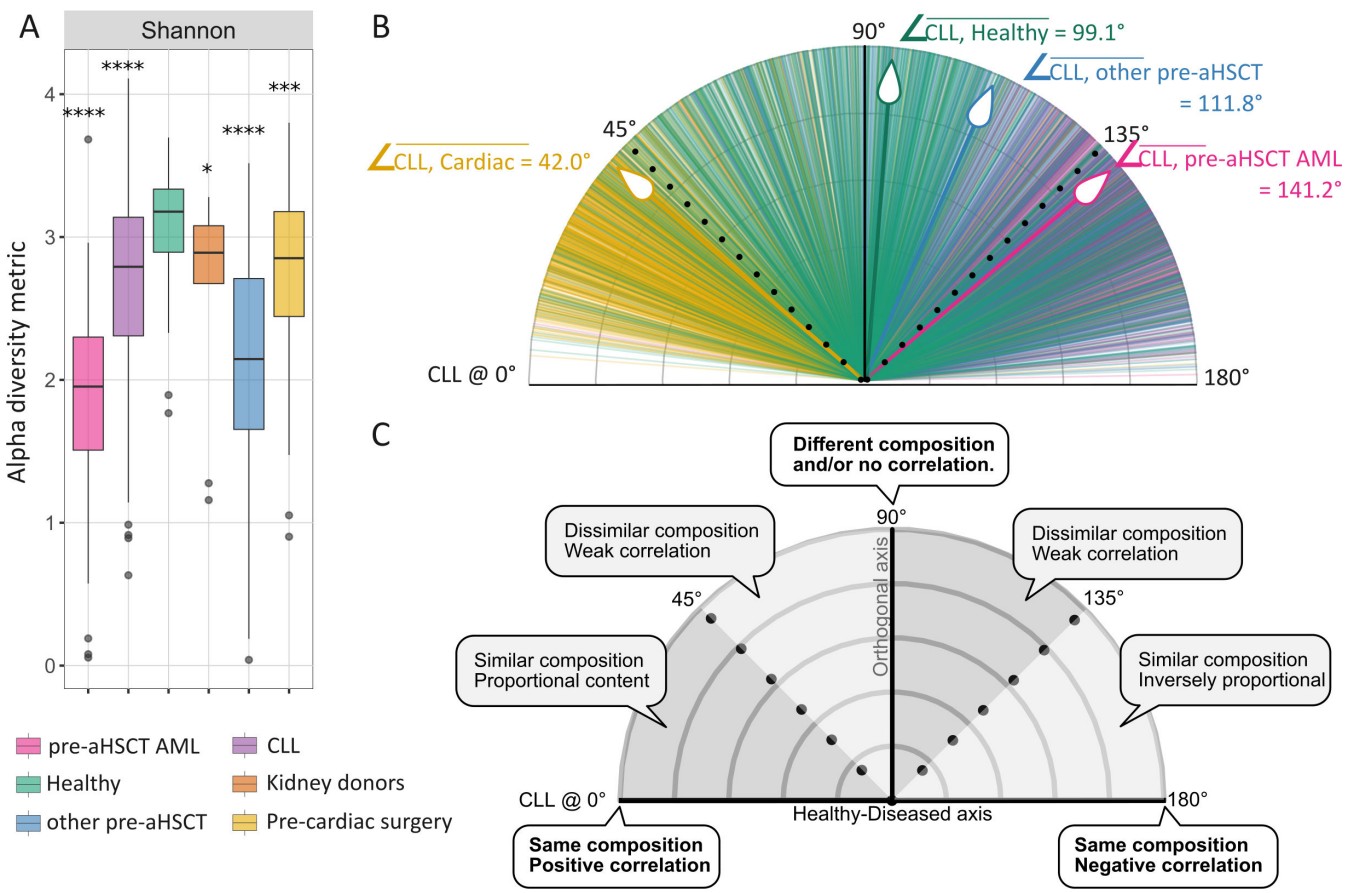

**FIG 1** Assessment of microbiome diversity and (dis)similarity across all included cohorts. (A) Alpha diversity measure (Shannon index) assessed at the species level. The statistical testing for significance uses healthy controls as the reference group (adjusted for multiple testing with BH, $P < 0.05$ was considered significant), and pairwise statistical testing results across all groups can be found in Table S2. Briefly, the diversity was not significantly different among CLL ($n = 85$), pre-cardiac surgery, and kidney donor cohorts. (B) Analysis of covariance by principal component analysis (PCA). To evaluate the similarity among the CLL microbiome and the other cohorts in a multidimensional space, a PCA was conducted. Angles between individual cohorts were calculated as the angles between vectors pointing to centroids of individual cohorts in PCA space (see Methods), with CLL being always positioned at 0°. The protractor-like plot represents all angles identified over 500 iterations between CLL and other individual cohorts. The thick lines with the white arrow-like symbols indicate the mean of centroid vectors per cohort. (C) The protractor-like plot provides interpretation of the angles based on cosine similarity principles, but arbitrarily applied to interpret biological differences between microbiome gut profiles. For instance, the other pre-aHSCT cohort has a dissimilar composition with the CLL cohort (given the SD = ±38.7° from the centroid), whereas the pre-aHSCT AML cohort is inversely correlated to CLL. In contrast, the pre-cardiac surgery cohort shows a similar composition in relation to CLL. *, $P \leq 0.05$; ***, $P \leq 0.001$; ****, $P \leq 0.0001$.

components identified as able to predict CLL were included (determined by GLM; $P < 0.05$; see Methods). Examining the relationship between all cohorts simultaneously, CLL gut profiles were inversely proportional (i.e., similar in composition but inverse in abundance) to both pre-aHSCT AML and other pre-aHSCT cohorts (∠CLL, pre-aHSCT AML = 141.2°± SD 22.5°, pink; ∠CLL, other pre-aHSCT = 111.8° ±SD 38.7°, blue). We observed a closer relationship between the CLL cohort and pre-cardiac surgery patients (∠CLL,CARDIAC = 42.0°± SD 27.7°, yellow) and a nearly orthogonal relationship (different composition and no correlation) between CLL and the healthy cohort (∠CLL, HEALTHY = 99.1° ±SD 40.8°, green). These results indicate that while CLL and pre-cardiac surgery patients have similar microbiome compositions, there is a strong inverse relationship between CLL and other hematological cohorts (please see the interpretation of angles in Fig. 1C).

## Assessing microbiome diversity by clustering of enterotypes

Based on the Calinski-Harabasz (CH) Index, the optimal number of clusters for partitioning the data on genus-level profiles was determined to be 2 (CH = 107.1), with five clusters being the second-best option (CH = 86.8). To explore the enterotypes reflecting the number of included cohorts, we opted for five clusters (Fig. 2A). Among the five clusters, four enterotypes driven by dominant bacterial genera such as *Bacteroides* (Bact 1 and 2), *Prevotella* (Pre), and *Ruminococcus* (Rum) were identified in our data (Fig. 2B). We identified an additional cluster (Cluster 2) characterized by the dominance of genera such as *Lachnoclostridium*, *Flavonifractor*, and *Escherichia*, often linked to diseased microbiome profiles (39, 40). The identification of this disease-associated enterotype was found almost solely (two CLL cases being the exception) in pre-aHSCT AML and other pre-aHSCT patients, our most diseased cohorts (Fig. 2B, Cluster 2). It is noteworthy that these samples form a distinct and compact cluster, suggesting a unique microbial community structure associated with a severe health condition and/or response to intensive antineoplastic and/or antimicrobial treatment.

## CLL can be better classified when the impact of medication data is eliminated

To validate and further refine the potential CLL microbiome pattern suggested in our previous studies (19, 29), we developed a machine learning classifier identifying CLL as an outcome including only CLL patients without recorded use of medication in 30 days and exposure to chemotherapy in 180 days prior to microbiome sampling ($n = 46$). The previously published cohort ($n = 10$ [29]), which served as the basis for our proposed CLL microbiome pattern, was not included in this validation analysis. Eliminating the impact of medication usage on the classification results allowed us to focus solely on identifying the disease-specific pattern. The groups of medication used as exclusion criteria are listed in Table 2. The comparator non-CLL group included only healthy individuals ($n = 59$) as both the pre-cardiac surgery cohort and kidney donors received medication. The results show that the three key features influencing the classification of CLL by CatBoost, also the best-performing model for this task, were the species *Eubacterium hallii*, *Dorea longicatena,* and *Anaerostipes hadrus*, all negatively correlated with the SHAP values (Fig. 3; Table S3). Three bacterial genera showing a positive correlation with SHAP values included *Alistipes shahii and Alistipes finegoldii* and *Eubacterium sp. CAG:180*, indicating that the probability of classifying CLL correctly increases as the relative abundance of these bacterial species increases. Interestingly, the performance of the CatBoost model to classify CLL without medication versus healthy controls was near perfect, with MCC = 0.98, PRAUC = 0.99, PPV = 0.98, and specificity = 0.98, and with only one misclassified sample across the fivefold cross-validation.

To get an insight into potential linear relationships between bacterial abundances and the outcome, we visualized the clr-transformed relative abundance of the top 10 important features (Fig. S1). We observed that CLL microbiomes were clearly depleted of the key species including *Eubacterium hallii*, *Dorea longicatena*, *Anaerostipes hadrus,* and *Coprococcus comes*. These results, together with the top features used for CLL classification, suggest a refined CLL-specific signature when compared to the healthy cohort.

In the second classifier, all CLL patients were compared to non-CLL patients from the pre-cardiac surgery, kidney donor (including those who had used medication within 30 days prior to sampling), and healthy cohorts. Pre-aHSCT AML and other pre-aHSCT cohorts were excluded as their microbiomes were strongly shaped by antimicrobials and chemotherapy and biased classification performance (Fig. S2A; Table S3). Incorporating medication groups (Table 2) as variables indicated that medication complicates classification by masking the disease-specific microbiome signature (Fig. S2B; Table S3).

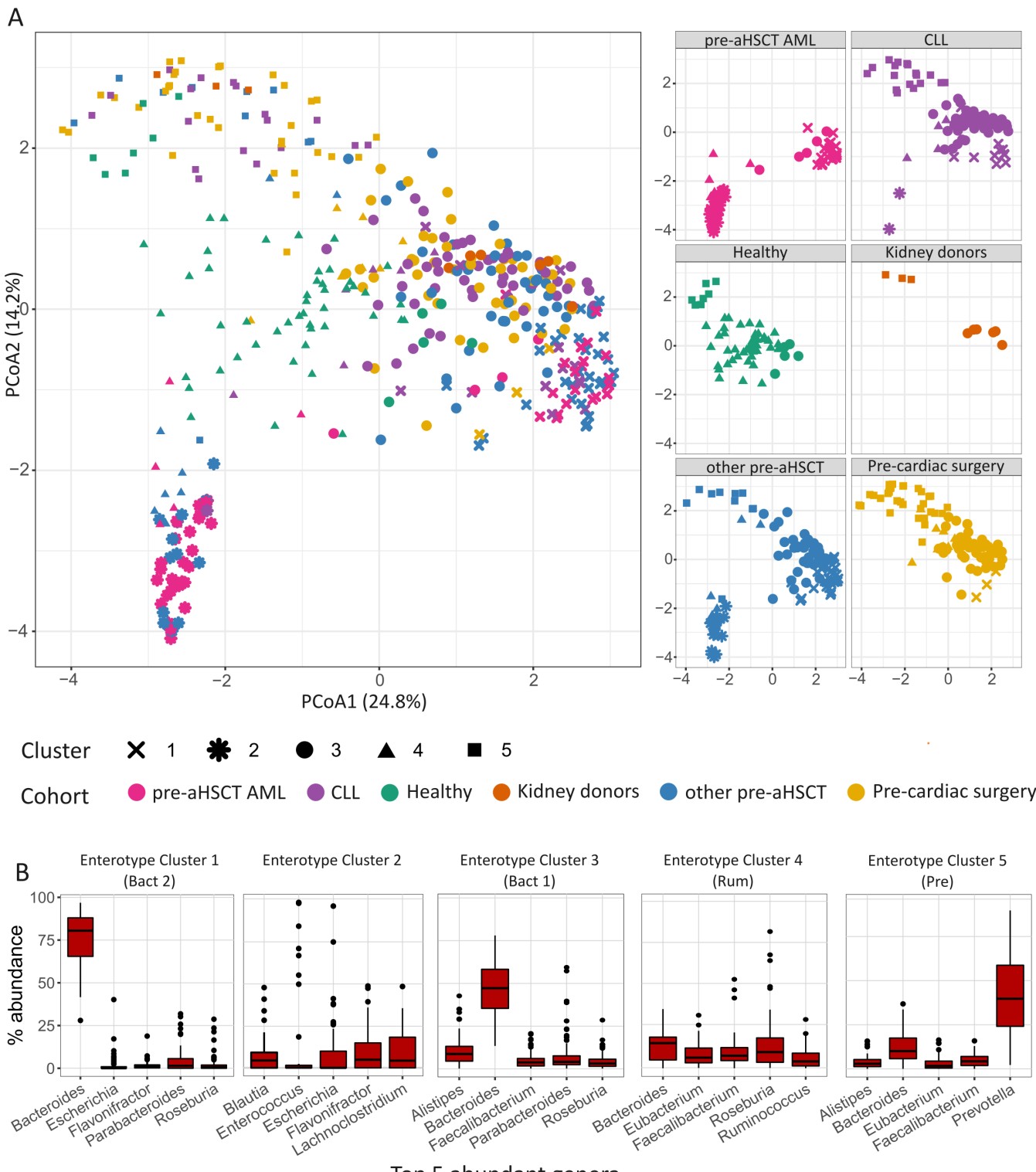

**FIG 2** Principal coordinate analysis (PCoA) plot of enterotype clusters and their composition. (A) PCoA clustering of microbial abundance data based on Jensen-Shannon divergence. Each point in the PCoA plot represents one sample colored by the cohort and shaped by the cluster number. Pre-aHSCT AML (pink), CLL ($n$ = 85, purple), healthy (green), kidney donors (orange), other pre-aHSCT (blue), and Pre-cardiac surgery (yellow). Panel to the left visualizes individual cohorts' distribution in the overall PCoA plot. Five partly overlapping clusters and one distinct cluster predominantly formed by clusters from pre-aHSCT AML and other pre-aHSCT cohorts were obtained for the included microbiome samples. (B) Box plots of the relative abundances (%) of the top five genera in each of the enterotype clusters. Clusters 1 and 3 are dominated by *Bacteroides*, whereas cluster 5 is dominated by the *Prevotella* genus. The top five genera in cluster 4 include, besides *Bacteroides*, mainly genera from *Firmicutes* phylum (*Roseburia, Ruminococcus, Faecalibacterium*, and *Eubacterium*).

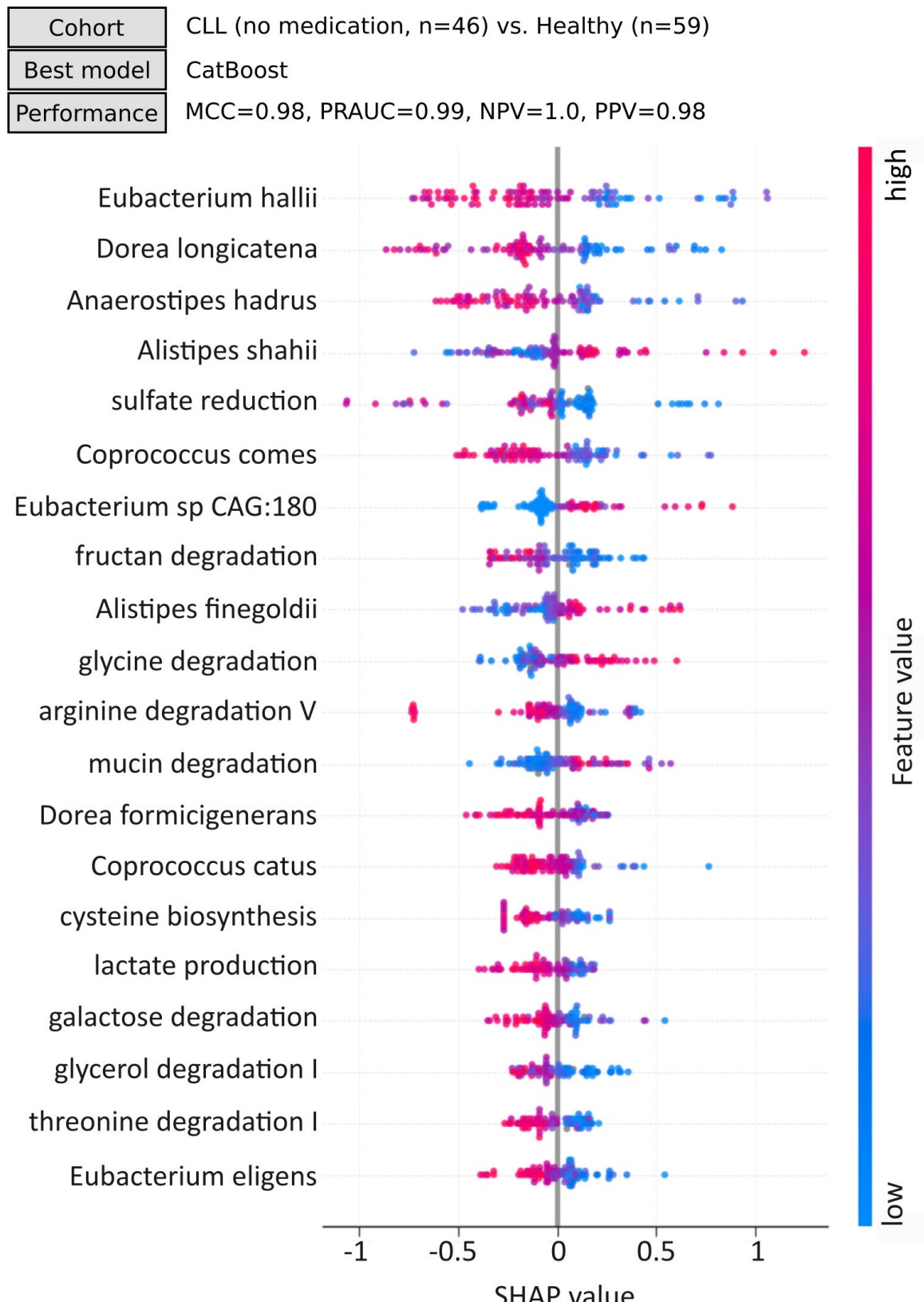

| Cohort | CLL (no medication, n=46) vs. Healthy (n=59) |
| --- | --- |
| Best model | CatBoost |
| Performance | MCC=0.98, PRAUC=0.99, NPV=1.0, PPV=0.98 |

**FIG 3** Machine learning (ML) model performance and outcome. A classification model was designed to classify CLL ($n$ = 46) as a binary outcome. CLL patients receiving any medication listed in Table 2 in 30 days prior to microbiome sampling were excluded. The previously published cohort ($n$ = 10 [29]), which served as the basis for our proposed CLL microbiome pattern, was not included in this validation analysis. The performance of the best-performing model was

Fig 3 (Continued)

calculated by taking the average of metrics across 5 folds. Metrics used for selection of the best performing classifier are listed. The SHapley Additive exPlanation (SHAP) summary plot illustrates the top 20 contributing bacterial species and GMM functional profiles to the identification of CLL using CatBoost as the best performing classifier. Each dot represents one patient and is colored such that red and blue represent higher and lower attribute value of a feature, respectively. Negative SHAP values associate with decreased predicted probabilities of CLL and vice versa. For instance, *Eubacterium hallii* had the highest contribution to the identification of CLL by CatBoost and was negatively correlated with its SHAP values, which translates to increased probability of identifying CLL as *Eubacterium hallii* decreases. *Alistipes shahii* was one of the features, with its values being positively correlated with SHAP values. In other words, the probability of identifying CLL increases as the relative abundance of *Alistipes shahii* increases. MCC, Matthews Correlation Coefficient (mean value across the fivefold validation; MCC value ranges from −1 to 1, 1 = perfect classification; 0 = random prediction); PRAUC, Precision-Recall Area Under Curve (summarizes trade-off between precision [positive predictive value] and recall [sensitivity]); PPV, positive predictive value (proportion of positive instances that were correctly classified); NPV, negative predictive value (proportion of negative instances that were correctly classified).

## Medication can explain as much variance in the gut microbiome as the underlying disease

To examine how much of the variance in the set of bacterial species and GMM functional profiles can be explained by microbiome metadata and the impact of medication groups, we employed redundancy analysis. The term variance here refers to the extent to which the composition of the different microbiome samples can be explained by the variables included in the analysis. For the first analysis, all cohorts except for pre-aHSCT AML and other pre-aHSCT (excluded due to chemotherapy regimens showing a strong impact on their classification) were included. Variables included disease (1 = CLL, pre-cardiac surgery cohort; 0 = healthy, kidney donors), use of any medication from Table 2 in 30 days prior to sampling (1/0), age, and sex. The results suggested that all variables combined could only explain 4.5% of the total variance of the metagenomic data at the species level and GMM functional profiles. Disease accounted for the largest variation in the gut microbiome, explaining 3.0% of the variance, followed by any medication explaining 1.3% of the variance (Fig. 4A).

Next, we only analyzed the diseased cohorts (CLL and pre-cardiac surgery patients) to be able to dissect the effect of individual medication groups and to observe the difference between the effect of two unrelated diseases compared to medication (Fig. 4B). Medication at the ATC 3rd level, type of disease, age, and sex combined explained 10.4% of the total variance. Within medication categories, corticosteroids and antibiotics explained 1.5% and 0.8% of the variance, respectively. Age and sex showed a consistent impact on the gut microbiome variance (0.9% and 0.8%, respectively) in both analyses (Fig. 4A and B). The percentage of variance explained by the disease variable in the cohort of CLL and pre-cardiac surgery patients dropped to 0.8%. The fact that disease alone can explain the same amount of variance in the microbiome as antibiotics suggests that medication can have an equally strong or even stronger (in the case of corticosteroids) impact on microbiome variance compared to the underlying disease itself, with the caveat kept in mind that correlations between disease and medication are evident (Table 2). Furthermore, the total variance explained by the combined effects of medication variables, age, and disease indicates that medication has little, but additive effect on top of disease phenotype for the gut microbiome variation.

Multivariate regression analysis, accounting for potential confounding variables such as age and gender, uncovered several significant associations between medication groups and bacterial species and GMM functional profiles (Fig. 4C). Out of the 11 drug classes (classified at the ATC 3rd level), five were significantly linked to at least one bacterial species (Table S4) when the combined effect of medication was considered. Interestingly, both antibiotics and metformin showed a strong association with *Enterococcus raffinosus*, a pathogen known for causing nosocomial infections due to its antibiotic resistance (41). Corticosteroids and metformin demonstrated the most significant associations with microbial species.

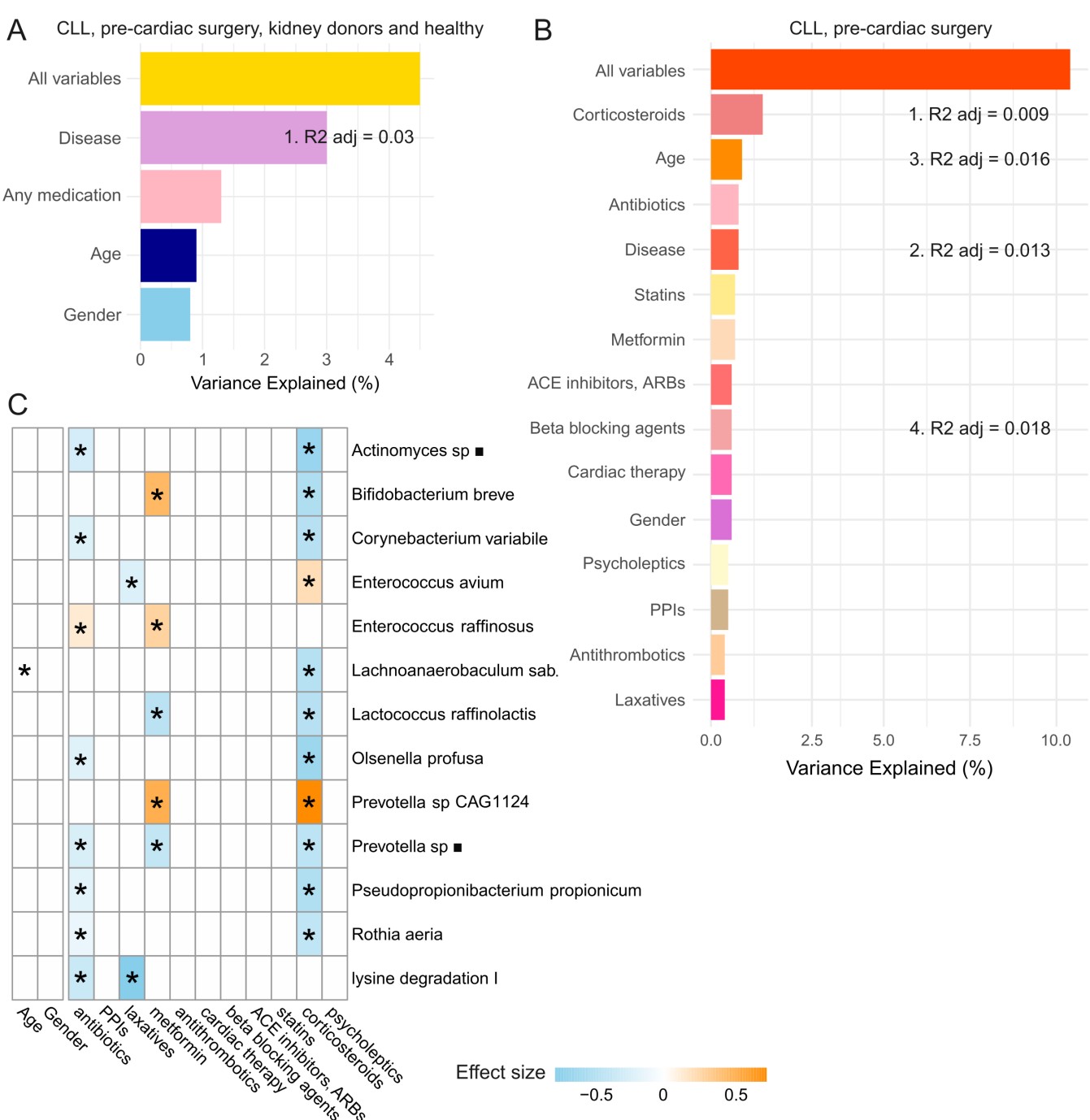

**FIG 4** Explained variance in the gut microbiome profiles. (A) Variance of the gut microbiome of patients with CLL, pre-cardiac surgery patients, kidney donors, and healthy individuals explained by the metadata categories. Any medication is a binary variable indicating any use of medication listed in Table 2 within 30 days prior to microbiome sampling. (B) Variance of the gut microbiome of patients with CLL and pre-cardiac surgery patients and kidney donors explained by the metadata and medication at the ATC 3rd level. Redundancy analysis was used for (A) and (B) variance analysis. Stepwise redundancy analysis applied to a model including all 11 medication groups, with only four variables selected for the final model that maximizes explanatory power while minimizing the complexity. The features selected in a stepwise redundancy analysis and their respective R2 adjusted values are displayed for the corresponding variables, with numbers 1–4 indicating the order of being added to the model. (C) Heatmap showing associations between the medication at the ATC 3rd level and bacterial species having at least two significant correlations with any variable assessed (all correlations can be found in Table S4). The associations were determined by multivariate regression analysis. Each cell is colored by effect sizes obtained from multivariate regression analysis, where orange and blue indicate positive and negative associations, respectively. Asterisks indicate significance ($P < 0.05$). No correction for multiple testing was performed as only variables with BH < 0.05 in the univariate regression analysis were included in the multivariate regression analysis. ACE inhibitors, angiotensin-converting enzyme inhibitors; ARBs, angiotensin receptor blockers; PPIs, proton pump inhibitors; *Lachnoanaerobaculum sab.*, *Lachnoanaerobaculum saburreum; square* indicates oral taxon.

## DISCUSSION

In this retrospective disease-entity study, we suggest a CLL gut microbiome signature and describe the impact of individual medication groups on the diseased gut microbiome. The key strength of the study is the inclusion of diverse cohorts, which ensures a comprehensive coverage of microbiomes associated with various hematological malignancies, other pathologies, and a healthy state. To our knowledge, this is the first study to integrate three layers of data—microbiome information, functional profiling, and medication data—along with comparisons across different cohorts in CLL microbiome research.

First, we revealed that the microbiome composition of CLL patients is closest to that of pre-cardiac surgery patients, whereas pre-aHSCT AML and other pre-aHSCT cohorts exhibited distinct differences from CLL in both alpha diversity and overall composition. Similar patterns were reported in pediatric patients, with ALL showing decreased alpha diversity at diagnosis and different compositions than healthy controls (42, 43). The differences in pre-aHSCT likely reflect the impact of high antimicrobial use and intense antineoplastic treatment, previously reported as detrimental to microbiome composition in aHSCT patients (28, 44, 45). These observations underscore the importance of accounting for the intensity of prior treatment and disease severity when interpreting microbiome data. The similarity between CLL and pre-cardiac surgery cohorts suggests that nonhematological conditions of comparable severity may shape the microbiome in similar ways. However, this resemblance may also be partly age-related, as previous studies in type 2 diabetes and colorectal cancer showed age explained more microbiome variation than the disease itself (46).

In the machine learning classification task to distinguish CLL from other cohorts, the inclusion of medication data was not beneficial. This suggests medication may homogenize microbiome composition across cohorts, thereby masking the disease-specific signals, consistent with antibiotics being selected as an important feature in the classifier that included medication. Our interpretation aligns with Bai et al., who reported that patients with ALL could not be distinguished from controls when both cohorts received antibiotics (47). The similarity in microbiome composition between CLL patients and the pre-cardiac surgery patients may also have influenced classification, complicating performance when medication was included and improving it when excluded. However, the near-perfect performance of the first classifier may also reflect the exclusion of the pre-cardiac surgery cohort, whose microbiome profiles closely resembled those of CLL patients, which could also be impacted by the long-term effect of antimicrobials more frequently used in the CLL cohort before and after diagnosis of CLL (14). In addition, as the first classifier compared CLL only to the healthy cohort, the identified pattern may reflect a general diseased profile rather than a CLL-specific profile. Therefore, validation in an external CLL cohort alongside both healthy and diseased individuals without medication use is required. Further, incorporating dietary information could as well improve classifications as diet is another major determinant of the microbiome (48).

As part of refining and validating the CLL-specific microbiome pattern, we identified *Eubacterium hallii*, *Dorea longicatena*, *Anaerostipes hadrus*, *Coprococcus comes*, and *Alistipes shahii, Alistipes finegoldii, Eubacterium sp. CAG:180* as the key CLL-associated features. In our previous study (29), we also identified Eubacterium, Dorea, Anaerostipes, and Coprococcus genera to be depleted as part of the proposed CLL microbiome signature when compared to another cohort of healthy controls. In our other study (19), both *Alistipes* species were identified as depleted in patients with the most severe course of CLL, which might imply their general protective role in the context of CLL. In accordance, *Alistipes shahii* was found to be enriched in healthy individuals when compared to patients with liver cirrhosis (49) and was associated with clinical remission of ulcerative colitis symptoms following fecal microbiota transplantation (50). *Alistipes finegoldii* has been studied for its role in gut dysbiosis and pro-inflammatory properties (51, 52) but also has been assigned anti-inflammatory properties as it seems to attenuate

the effects of colitis (53). *Eubacterium sp. CAG:180* was positively associated with plasma levels of inflammation-related proteins in obesity (54). As *Eubacterium* species are members of the gut microbiota generally known for their role in producing SCFAs like butyrate, which are crucial for colon health (55), further research is required to fully understand the association with disease. It is crucial to note that the suggested signature is based on the SHAP values and their correlation with the predicted outcome. Therefore, the signature reflects not only the effect of individual features but also their combined impact with likely complex relationships. Importantly, it remains unclear whether this microbiome signature is a consequence of CLL or whether it contributes to disease development, which should be acknowledged as a limitation. While our previous mouse model study provides initial evidence supporting the causal role of the microbiome in CLL (19), more prospective studies are required to clarify the causal direction of these associations.

Finally, we demonstrated that medication influences microbiome variance in a detectable way, though to a lesser extent than expected. Among the medications, corticosteroids and antibiotics had the most significant impact on gut microbiome variation. When the effect of medications on microbiome variation was assessed along with age, sex, and disease variables, the combined factors explained 10.4% of the total variance in CLL and pre-cardiac surgery patient groups (representing medicated hematological and nonhematological conditions). It is in line with a study by Falony et al. reporting 10% of microbiome variance explained by medication and around 16% explained when other factors such as diet, disease, blood parameters, or anthropometrics were included (56). A study in the Dutch population reported that approximately 18% of the microbiome variance can be explained by 126 factors (57). A Japanese population-wide study reported medication to account for 7% of the microbiome variance at genus level, followed by disease and anthropometrics, among other factors (35). Although the findings underline the importance of accounting for a wide range of factors in the future microbiome studies, comparing results across studies is challenging. For example, the way diet is analyzed as a factor can vary significantly. In the study by Zhernakova et al. (57), 60 dietary factors were examined, whereas the study by Nagata et al. (35) included 23 factors.

The significant increase in the abundance of *Enterococcus* species associated with the use of antibiotics, corticosteroids, and metformin in this study aligns with previous studies reporting an increase in Enterococcus levels following antibiotic use and administration of PPIs (58–61). In our study, the effect of some drugs might have been missed due to small sample sizes and thus small numbers of individuals using different medications. Additionally, the complex interactions resulting from polypharmacy or excessive medication use remain poorly understood and are challenging to interpret in terms of their impact on the gut microbiome. Future research with larger sample sizes and a focus on drug-microbiome interactions could provide clearer insights (62, 63).

Although the sample size in our study is comparable or larger than in other studies in CLL (64, 65) and other hematological malignancies with a microbiome perspective (42, 43, 66), it may impact the generalizability of our findings. Other limitations are the lack of one-to-one matched healthy controls for CLL patients in terms of age and sex. In addition, while the inclusion of multiple cohorts enhances the robustness of our study, it also introduces variability due to unbalanced medication usage. The varying medication regimens across cohorts are an inherent problem as more severe cases typically involve more extensive medication. Although the ML models have been designed to address imbalanced class distributions using sample weights, they might not entirely eliminate the potential for bias introduced by different medication practices. Lastly, our study did not include external validation, which limits the generalizability of our findings across different populations representing dietary and genetic diversity, previously shown to impact the microbiome composition (48, 57, 67–69).

In conclusion, we demonstrate the potential of using microbiome data to characterize disease states such as CLL. The study refined the previously identified microbiome

patterns associated with CLL and quantified the impact of different medication usage. Several bacterial species were positively or negatively correlated with predicted probabilities of CLL, highlighting potential targets for microbiome modulation via introduction of beneficial bacteria and/or elimination of pathogens. Despite microbiome signatures reflecting disease categories only to some degree, and while medication patterns seem to blur such patterns, ongoing studies with time-series data, i.e., samples collected before and after drug intake, will allow us to understand the dynamics of the gut microbiome upon CLL treatment. As a future perspective, longitudinal microbiome analyses of patients with CLL might also be of value across different phases of the disease, including Richter transformation (70).

## ACKNOWLEDGMENTS

The authors would like to sincerely thank the patients who provided samples; their contributions were vital to making this research possible. We would also like to acknowledge the PERSIMUNE Centre of Excellence for its infrastructure, financial support (Grant 126), and expertise, all of which were integral to the study. Furthermore, we recognize the efforts of the trial unit staff at the Department of Hematology and the PERSIMUNE biobank at Rigshospitalet for their invaluable role in organizing and collecting the samples.

E.E.I. is supported by the BRIDGE – Translational Excellence Programme (bridge.ku.dk) at the Faculty of Health and Medical Sciences, University of Copenhagen, funded by the Novo Nordisk Foundation. Grant agreement no. NNF20SA0064340 and Grant number NNF18CC0034900. C.U.N. is funded by the Alfred Benzon Foundation and Novo Nordisk Foundation.

C.U.N., T.F.K., and R.Z.M. conceived the study. T.F.K. collected and analyzed the data. R.Z.M. conducted the machine learning analyses and gave input on the statistical approach. C.B. contributed to establishing a data source of CLL clinical data. A.G. conducted the taxonomical profiling and gut functional profiling analyses and gave input on statistical methods. E.E.I., H.S., C.B., S.R., and C.Z. contributed to interpretation of the medication and clinical data. C.U.N., C.D.C.-B., H.S., E.E.I., H.B.R., R.V.N., C.Z., and S.S.S. included patients in the study and collected patient samples. T.F.K., E.E.I., and C.U.N. wrote the draft paper, and all authors contributed to the paper and reviewed the final version.

## AUTHOR AFFILIATIONS

[1]Department of Hematology, Rigshospitalet, Copenhagen, Denmark
[2]Danish Cancer Institute, Copenhagen, Denmark
[3]Centre of Excellence for Health, Immunity and Infections (CHIP), Rigshospitalet, Copenhagen University Hospital, Copenhagen, Denmark
[4]Novo Nordisk Foundation Center for Basic Metabolic Research, Faculty of Health and Medical Sciences, University of Copenhagen, Copenhagen, Denmark
[5]Department of Cardiothoracic Surgery, Rigshospitalet, Copenhagen, Denmark
[6]Department of Anesthesiology, Odense University Hospital, Odense, Denmark
[7]Department of Intensive Care, Odense University Hospital, Odense, Denmark
[8]Department of Cardiology, Rigshospitalet, Copenhagen, Denmark
[9]Novo Nordisk Foundation, Hellerup, Denmark
[10]Department of Cardiothoracic Anesthesiology, Rigshospitalet, Copenhagen, Denmark
[11]Department of Nephrology, Rigshospitalet, Copenhagen, Denmark
[12]Department of Clinical Medicine, University of Copenhagen, Copenhagen, Denmark

## AUTHOR ORCIDs

Tereza Fait Kadlec  http://orcid.org/0000-0003-2729-0354
Ramtin Zargari Marandi  http://orcid.org/0000-0001-9233-1656
Carsten Utoft Niemann  http://orcid.org/0000-0001-9880-5242

## AUTHOR CONTRIBUTIONS

Tereza Fait Kadlec, Conceptualization, Data curation, Formal analysis, Investigation, Methodology, Validation, Visualization, Writing – original draft, Writing – review and editing | Carsten Utoft Niemann, Conceptualization, Funding acquisition, Project administration, Resources, Supervision, Writing – review and editing.

## DATA AVAILABILITY

The data sets generated and analyzed during this study are derived from patients treated in Denmark. The data sets contain sensitive patient data governed by GDPR and Danish law. Due to Danish legislation (Act No. 502 of 23 May 2018) and approvals granted by the Danish Data Protection Agency, and due to the individual patient-level data that cannot be anonymized, only pseudonymized, it is not possible to upload data to a publicly available database. However, data including processed data and code are readily available in an in-house database and can be accessed upon reasonable request to the corresponding author, provided that a data transfer agreement is established in accordance with current regulations.

## ETHICS APPROVAL

The project was performed in compliance with the guidelines of the Declaration of Helsinki and approved by the Danish national ethics committee (Approval No. 1804410, Journal No. H-16047481), as well as the Danish Data Protection Agency (RH-2017-67, I-suite 05320). Written informed consent was obtained from all patients prior to sampling.

## ADDITIONAL FILES

The following material is available online.

### Supplemental Material

**Supplemental Material (Spectrum00944-25-s0001.pdf).** Supplemental methods; Fig. S1 and S2.
**Table S1 (Spectrum00944-25-s0002.xlsx).** CLL patient characteristics, PERMANOVA testing, and detailed diagnoses of other pre-aHSCT cohort.
**Table S2 (Spectrum00944-25-s0003.xlsx).** Alpha diversity statistics.
**Table S3 (Spectrum00944-25-s0004.xlsx).** Machine learning results.
**Table S4 (Spectrum00944-25-s0005.xlsx).** Multivariate regression results.

### Open Peer Review

**PEER REVIEW HISTORY (review-history.pdf).** An accounting of the reviewer comments and feedback.

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
