## [Reviewer comments · Microbiology Spectrum]

Microbiology Spectrum

Explainable Machine Learning to Identify Chronic Lymphocytic Leukemia and Medication Use Based on Gut Microbiome Data

Tereza Fait Kadlec, Emma Ilett, Caspar da Cunha-Bang, Henrik Sengeløv, Christian Brieghel, Arda Gulay, Sulman Rafiq, Hanne Ravn, Chaoqun Zheng, Rikke Nielsen, Søren Sørensen, Ramtin Marandi, and Carsten Niemann

Corresponding Author(s): Carsten Niemann, Rigshospitalet

Review Timeline:

Submission Date:	April 19, 2025
Editorial Decision:	August 5, 2025
Revision Received:	September 24, 2025
Accepted:	October 5, 2025

Editor: S. Wesley Long

Reviewer(s): The reviewers have opted to remain anonymous.

Transaction Report:

DOI: <https://doi.org/10.1128/spectrum.00944-25>

Re: Spectrum00944-25 (Explainable Machine Learning to Identify Chronic Lymphocytic Leukemia and Medication Use Based on Gut Microbiome Data)

Dear Dr. Carsten Utoft Niemann:

Thank you for the privilege of reviewing your work. Below you will find my comments, instructions from the Spectrum editorial office, and the reviewer comments.

Please fully address the reviewers comments in your resubmission.

Revision Guidelines

Sincerely,
S. Wesley Long
Editor
Microbiology Spectrum

Reviewer #1 (Comments for the Author):

In this study, Faitová et al. used a machine learning model to integrate clinical, metagenomic and medication data to assess the gut microbiome signature in 85 patients affected by chronic lymphocytic leukemia (CLL), that represents the most frequent leukemia in the adult population. Data were compared to hospital controls and to healthy individuals. Interestingly, the results document that up to 95% of microbiome variation remained unaccounted by the factors included in the analysis. Surprisingly, the

microbiomes of CLL patients displayed a high degree of similarity with that of patients undergoing cardiac surgery, while instead patients receiving intensive cancer treatment had the least diverse microbiomes. The study is well designed and well written. Overall, these findings prompt new avenues of microbiota research in CLL and provide pivotal evidence of the relevance of microbiota variation in this disease. A few issues require further care for improvement.

MAJOR ISSUES

1. The authors correctly acknowledge in the Introduction that CLL patients frequently exhibit compromised immune systems leading to immunodeficiency and predisposition to infections. This immunological condition may affect the gut microbiota. Which percentage of the CLL patients of this cohort were receiving IVIGs? Did IVIG therapy correlate with any specificities of the gut microbiota?
2. The authors compare the CLL microbiota with that of other clinical conditions, namely pre-allogeneic HSCT acute myeloid leukemia and myelodysplastic syndromes and pre-cardiac surgery patients. Although such comparisons are interesting and revealing, one should consider that CLL per se is a very heterogeneous disease. Was there any association between CLL clinical (treatment naive, 1st line, relapsed/refractory) and/or genetic biomarkers with specific features of the gut microbiota?
3. On page 9, the authors state that they refined their analysis by "including only CLL patients without recorded use of medication in 30 days and exposure to chemotherapy in 180 days prior to microbiome sampling". Since CLL is now treated with chemo-free strategies, the authors should discuss whether any microbiota differences emerged when comparing patients receiving/treated with chemotherapy versus patients receiving/treated with pathway inhibitors.
4. In the Discussion, the authors should briefly mention as a future perspective that longitudinal microbiota analysis of CLL patients might be of value across different phases of the disease, including Richter transformation (Maher et al., *Molecular Mechanisms in the Transformation from Indolent to Aggressive B Cell Malignancies*. *Cancers (Basel)*. 2025 17(5):907).

MINOR ISSUES

1. The manuscript may benefit from a certain (5%) reduction in length.

Reviewer #2 (Comments for the Author):

Additional statistical test should be provided

The goals of this study are to assess whether microbial signatures in CLL exist, whether hematological signatures exist, in general, and the impact of medication on these signatures.

It's a potentially interesting study of microbiome and medication relationship, but hard to draw conclusions from this work. We do not have details about how the fecal samples were collected relative to timing of diagnosis and how representative the cohort is. Even for CLL, which is a main purpose of the study, details of the CLL patients are lacking (e.g., stage, IGHV status, treatment status). The hematological malignancies cohort is also lacking details and not very representative, but rather focused on those with aggressive diseases that need HSCT. It is unknown how even the pre-surgical cardiac cohort is representative and why included. Taken together, this study is a collection of individuals who gave a fecal sample, but making conclusions needs to be tempered until more details about the individuals and more justification of the study design are provided. Finally, the CLL signature seems to be an extension of prior work. The authors should be more clear as to what is novel herein.

Below are more specific comments.

Abstract is hard to follow and poorly written. One has to read the paper to understand what the abstract is saying. Examples of issues are below.

- The authors state "other hospital cohorts (N=263)". What does this mean, how many cohorts. One does not know any of this until you read the manuscript.
- The next sentence uses "patients prior to elective cardiac surgery". How many individuals is this? Is it all 263 individuals? Then the next sentence goes into, "patients prior to planned aHSCT", how many individuals? When you say "all included cohort", is that all 263 hospital cohorts? Does this include the healthy individuals?
- The next sentence states, "Using a machine learning approach", is this ML approach different from that used in the prior sentences? The rest of this sentence seems to conflict with an earlier written sentence "our analyses revealed similarities in both diversity and composition between microbiomes of pts with CLL and pts of elective cardiac surgery. It seems the reference group is different depending on which group is used to compare to CLL. Why are you comparing to cardiac pts in the first place? They seem different than healthy controls
- The sentence, "while we demonstrated that medication and underlying disease had a significant influence on ...". This was not demonstrated in the abstract. The prior sentence, "when addressing microbiome variation, medication disease, etc", isn't clear and doesn't support the sentence, "while we demonstrated that ...".

The text for Importance is written better but with more clarity still needed:

1. The sentence, "Pts receiving intensive cancer treatment had least diverse microbiomes". Who are these Pts. CLL? Other cancers?
2. The sentence, "Using machine learning, this ..." didn't you use machine learning to support the prior sentence?
3. "The findings contribute to microbial biology by expanding...", Is this microbial biology of CLL pts or in general?
4. When using the word Patients, it's not clear if referring to CLL patients or some other patients. Please be more specific.

Please clarify when the fecal samples were collected in the CLL patients and other patients. In particular, where samples collected after respective cancer treatment.

Why was medication collected only between 30 days and 1 day prior to fecal sample date? Is there data to support that the microbiome recovers after 30 days of medication?

104 individuals with pre aHSCT are MDS or "other hematological diseases". What are the other hematological diseases? Myeloid? Lymphoid? How representative are these individuals with hematological malignancies?

Why are the 9 kidney donors a separate cohort? Why are they not combined with healthy controls?

The basis for including patients pre cardiac surgery is not justified. Why does it matter that CLL patients have microbiome similar to pre-cardiac surgery patients. The meaning of this is unclear.

Table 2 is not adding much value. It's showing raw counts of medication usage across medication types and cohorts. It should be a supplemental table. I would add a row indicating no medications. This is because it seems to be implied that everyone in the pre-cardiac or kidney donor cohorts had a least one medication (based on Figure 3). One cannot tell this information based on data in Table 2.

Figure 1.A: Please add in statistical tests comparing between the cohorts. I suggest using the healthy controls as the reference group.

Figure 1B: interesting figure but not sure if these differences in angle are statistically different from CLL (0 degrees) given the high SD values.

Figure 3: Why compare CLL to the combined non-cancer and healthy controls. The basis of this comparison is not justified.

The second part of Figure 3 (CLL vs healthy controls) is not novel, per se, but rather an extension of the authors' prior work. Thus, the Discussion should be reduced or clarified as to what is gained by these additional analyses. Also it is unknown whether

the signature is driven by the CLL disease or the signature drives the disease. Should be stated in limitations and need for future prospective studies.

Point by point Responses to the Reviewer's comments

Thank you very much for the review of our manuscript entitled: "Explainable Machine Learning to Identify Chronic Lymphocytic Leukemia and Medication Use Based on Gut Microbiome Data".

We appreciate all the valuable comments and suggestions, which we believe helped us to improve the quality of the article. Our responses to the Reviewers' comments are listed below in a point-by-point manner. Appropriate changes, suggested by the Reviewers, have been introduced to the manuscript (highlighted in this document where applicable and in the manuscript by using track changes).

Reviewer #1 (Comments for the Author):

In this study, Faitová et al. used a machine learning model to integrate clinical, metagenomic and medication data to assess the gut microbiome signature in 85 patients affected by chronic lymphocytic leukemia (CLL), that represents the most frequent leukemia in the adult population. Data were compared to hospital controls and to healthy individuals. Interestingly, the results document that up to 95% of microbiome variation remained unaccounted by the factors included in the analysis. Surprisingly, the microbiomes of CLL patients displayed a high degree of similarity with that of patients undergoing cardiac surgery, while instead patients receiving intensive cancer treatment had the least diverse microbiomes. The study is well designed and well written. Overall, these findings prompt new avenues of microbiota research in CLL and provide pivotal evidence of the relevance of microbiota variation in this disease. A few issues require further care for improvement.

MAJOR ISSUES

1. The authors correctly acknowledge in the Introduction that CLL patients frequently exhibit compromised immune systems leading to immunodeficiency and predisposition to infections. This immunological condition may affect the gut microbiota. Which percentage of the CLL patients of this cohort were receiving IVIGs? Did IVIG therapy correlate with any specificities of the gut microbiota?

We appreciate this important suggestion. After reviewing electronic health records for all 85 CLL patients, we found that 7 patients (8.2%) had received IVIG before microbiome sampling. Using PERMANOVA (adonis2() in the vegan R package), we observed no significant correlation between IVIG therapy and microbiome composition. However, we note that the limited number of IVIG-treated patients may have reduced statistical power. These results are now included in Supplemental Table 1.

Statistical testing results

Variable	Df	SumOfSqs	R2	F	Pr(>F)
Treatment naive at sampling	1	0,3021	0,01211	1,0173	0,414
Treatment type	2	0,4124	0,06844	0,6612	0,962
Immunoglobulin replacement therapy	1	0,2758	0,01105	0,9276	0,545

IGHV mutation	1	0,2801	0,01177	0,9289	0,534
TP53 mutation status	1	0,2539	0,01163	0,8742	0,64

Formula used: `adonis2(bray_curtis_distance_matrix ~ Variable, data, na.omit = T)`

We also updated the Results sections so that it's apparent to the reader:

...As anticipated, exposure to both chemotherapy and antibiotics was gradually higher among patients with CLL, other other pre-aHSCT, and prepre-aHSCT AML (Chi2 between-group difference: CLL vs pre-aHSCT AML, $P=0.01$; CLL vs other pre-aHSCT, $P=0.02$). In the CLL group, no significant difference was observed in microbiome composition with respect to treatment status, immunoglobulin replacement therapy, or genetic biomarkers such as IGHV mutation status or TP53 mutation (Supplemental Table 1).

2. The authors compare the CLL microbiota with that of other clinical conditions, namely pre-allogeneic HSCT acute myeloid leukemia and myelodysplastic syndromes and pre-cardiac surgery patients. Although such comparisons are interesting and revealing, one should consider that CLL per se is a very heterogeneous disease. Was there any association between CLL clinical (treatment naive, 1st line, relapsed/refractory) and/or genetic biomarkers with specific features of the gut microbiota?

We appreciate your observation about the need for further address heterogeneity within the CLL cohort and testing to rule out potential biases. We evaluated the relationship between microbiome composition and available clinical/genetic markers, including treatment status (treatment-naïve, 1st line and relapsed/refractory, $p = 0.41$), IGHV mutation ($p = 0.53$), and TP53 mutation ($p = 0.64$). None of these variables significantly correlated with microbial community composition. PERMANOVA tests were applied (`adonis2()` in `vegan` R package) and results are presented in Supplemental Table 1.

Statistical testing results

Variable	Df	SumOfSqs	R2	F	Pr(>F)
Treatment naive at sampling	1	0,3021	0,01211	1,0173	0,414
Treatment type	2	0,4124	0,06844	0,6612	0,962
Immunoglobulin replacement therapy	1	0,2758	0,01105	0,9276	0,545
IGHV mutation	1	0,2801	0,01177	0,9289	0,534
TP53 mutation status	1	0,2539	0,01163	0,8742	0,64

Formula used: `adonis2(bray_curtis_distance_matrix ~ Variable, data, na.omit = T)`

We also updated the Results sections so that it's apparent to the reader:

...As anticipated, exposure to both chemotherapy and antibiotics was gradually higher among patients with CLL, other pre-aHSCT, and prepre-aHSCT AML (Chi2 between-group difference: CLL vs pre-aHSCT AML, $P=0.01$; CLL vs other pre-aHSCT, $P=0.02$). In the CLL group, no significant difference was observed in microbiome composition with respect to treatment status, immunoglobulin replacement therapy, or genetic biomarkers such as IGHV mutation status or TP53 mutation (Supplemental Table 1).

3. On page 9, the authors state that they refined their analysis by "including only CLL patients without recorded use of medication in 30 days and exposure to chemotherapy in 180 days prior to microbiome sampling". Since CLL is now treated with chemo-free strategies, the authors should discuss whether any microbiota differences emerged when comparing patients receiving/treated with chemotherapy versus patients receiving/treated with pathway inhibitors.

We agree this is an important consideration. Our cohort reflect representative CLL population in terms of treatment need, Binet stage and other factors. We tested whether treatment modality (chemotherapy vs. pathway inhibitors) was associated with microbiome composition. No significant associations were detected, although limited subgroup sizes likely reduce statistical power ($p = 0.96$). Results are included in the new Supplemental Table 1.

```
adonis2(formula = bray_curtis_dist ~ as.factor(Treat_type), data = CLL_taxonomical, na.action = na.omit)
      Df SumOfSqs      R2      F Pr(>F)
Model   2   0.4124 0.06844 0.6612  0.962
Residual 18   5.6129 0.93156
Total   20   6.0253 1.00000
```

In the machine learning analysis designed to validate CLL microbiome pattern, we restricted the dataset to 46 CLL patients (excluding those with recent medication use or chemotherapy). After further review, we found that only 4 patients (treated with pathway inhibitors) were included in this analysis; all other patients who had received any treatment for CLL prior to microbiome sampling were excluded by the medication use criteria.

4. In the Discussion, the authors should briefly mention as a future perspective that longitudinal microbiota analysis of CLL patients might be of value across different phases of the disease, including Richter transformation (Maher et al., Molecular Mechanisms in the Transformation from Indolent to Aggressive B Cell Malignancies. *Cancers (Basel)*. 2025 17(5):907).

We agree and have revised the Discussion accordingly. We now state that:

... Despite microbiome signatures reflecting disease categories only to some degree, and while medication patterns seem to blur such patterns, ongoing studies with time-series data, i.e. samples collected before and after drug intake, will allow us to understand the dynamics of the gut microbiome upon CLL treatment. As a future perspective, longitudinal microbiome analyses of patients with CLL might also be of value across different phases of the disease, including Richter transformation (71).

MINOR ISSUES

1. The manuscript may benefit from a certain (5%) reduction in length.

Thank you for pointing this out, we agree, and have thus revised and reduced its length by ~10%

(from 4,850 to ~4,350 words), while maintaining the main messages and improving clarity.

Reviewer #2 (Comments for the Author):

The goals of this study are to assess whether microbial signatures in CLL exist, whether hematological signatures exist, in general, and the impact of medication on these signatures. It's a potentially interesting study of microbiome and medication relationship, but hard to draw conclusions from this work.

We do not have details about how the fecal samples were collected relative to timing of diagnosis and how representative the cohort is. Even for CLL, which is a main purpose of the study, details of the CLL patients are lacking (e.g., stage, IGHV status, treatment status). The hematological malignancies cohort is also lacking details and not very representative, but rather focused on those with aggressive diseases that need HSCT. It is unknown how even the pre-surgical cardiac cohort is representative and why included. Taken together, this study is a collection of individuals who gave a fecal sample, but making conclusions needs to be tempered until more details about the individuals and more justification of the study design are provided.

We appreciate this constructive feedback, and we understand the concern that our manuscript did not provide clear enough details to explain the study design.

To address this, we have now clarified in the revised manuscript that all samples were collected under a standardized sampling protocol within the framework of the PERSIMUNE microbiome sampling effort, thus not being random fecal samples compiled into one paper. The protocol included parallel projects taking place at Copenhagen University Hospital and included three hospital cohorts: (1) CLL patients, (2) patients undergoing allogeneic HSCT, and (3) pre-cardiac surgery patients. We have added a citation to the PERSIMUNE, Copenhagen University Hospital framework to document and explain this approach. The overall aim of PERSIMUNE was to address immune deficiency across different diseases.

For the pre-cardiac cohort, we have clarified that these samples were collected before surgery to provide a baseline sample. This group was included as a representation of critical illness unrelated to hematological malignancy serving as a contrast cohort, addressing whether critical illness (with unrelated pathogenesis) could identify a "critical illness" signature rather than a disease specific microbiome signature.

In addition, we have added the requested clinical details for the CLL cohort, including TP53 mutation status, IGHV mutation status, and treatment status. These data are now presented in Supplemental Table 1, and we have performed additional statistical testing as suggested by the first reviewer. We believe this additional detail provides a much clearer picture of how representative our CLL cohort is and how the samples relate to disease stage and treatment.

Feature		All (n=85)
Time from diagnosis to sampling	years (median, IQR)	1.9 (0.3, 5.3)
Binet stage at diagnosis	A	72 (85%)
	B	10 (12%)
	C	3 (3%)
Treatment status at sampling	Treatment naive	64 (75%)
	Treated	18 (21%)
	Relapse-refractory	3 (4%)
Treatment type*	Chemotherapy (FCR, RCHOP+HD-MTX)	12 (14%)
	Chemoimmunotherapy (R+Bendamustin)	4 (5%)
	Targeted therapy**	5 (7%)
Immunoglobulin treatment before microbiome sampling	Treated	7 (8%)
	Not treated	78 (92%)
IGHV	M-CLL	44 (52%)
	U-CLL	36 (42%)
	NA	5 (6%)
TP53	Mutated	10 (12%)
	Unmutated	65 (76%)

Finally, the CLL signature seems to be an extension of prior work. The authors should be more clear as to what is novel herein.

We sincerely appreciate and acknowledge that our findings on the CLL-associated microbial signature represent an extension of prior work. We agree that it is essential to clearly distinguish what is novel in this study. But at the same time, we feel it is important for the scientific community that previous findings are validated or refined as more samples become available as this supports the robustness.

To address the reviewer's concern, we have substantially trimmed the sections of the manuscript that focused on the validation of our earlier work. By doing so, we hope to avoid overemphasizing what is confirmatory and instead draw the reader's attention to the medication-related analyses, which we believe represent the most novel contribution of this study.

We also now clearly state that the previously published cohort (n = 10; (29)), which served as the basis for our proposed CLL microbiome pattern, was not included in this validation analysis.

Table 1 legend: ... Eight-five patients with CLL included 70 previously published (19,29). The previously published cohort of 10 patients (29) which served as the basis for our proposed CLL microbiome pattern, was excluded in microbiome pattern validation analysis (Figure 3).

Reviewed Results section: To validate and further refine the potential CLL microbiome pattern suggested in our previous study (24), we developed a machine learning classifier identifying CLL as an outcome including only CLL patients without recorded use of medication in 30 days and exposure to chemotherapy in 180 days prior to microbiome sampling (n=46). The previously

published cohort (n = 10; (24)), which served as the basis for our proposed CLL microbiome pattern, was not included in this validation analysis.

Below are more specific comments.

Abstract is hard to follow and poorly written. One has to read the paper to understand what the abstract is saying. Examples of issues are below.

The authors state “other hospital cohorts (N=263)”. What does this mean, how many cohorts. One does not know any of this until you read the manuscript.

The next sentence uses “patients prior to elective cardiac surgery”. How many individuals is this? Is it all 263 individuals? Then the next sentence goes into, “patients prior to planned aHSCT”, how many individuals? When you say “all included cohort”, is that all 263 hospital cohorts? Does this include the healthy individuals?

Thank you for pointing out that we had to make the abstract clearer and more readable, we have updated it and the included cohorts as well as their sizes have been detailed in the abstract

Abstract

Medication, particularly antibiotics, significantly alters gut microbiome composition, often reducing microbial diversity and affecting host health. Given that the gut microbiome may influence cancer progression, we integrated clinical, shotgun metagenomic, and medication data ~~using machine learning models~~ to assess microbiome composition across diseased and healthy cohorts, as well as the impact of medication on microbiome variation, the potential gut microbiome signature in patients with chronic lymphocytic leukemia (CLL; N=85), compared to other hospitalThe study cohorts included patients with chronic lymphocytic leukemia (CLL, n=85), acute myeloid leukemia (AML, n=61), myeloid dysplastic syndrome (MDS) and other severe hematological malignancies (n=104), patients scheduled for elective cardiac surgery (n=89), and kidney donors (n=9), all collected as part of a consecutive microbiome sampling effort at Copenhagen University Hospital, Denmark, (N=263) and healthy individuals (N=59). ~~First, our analyses~~

The next sentence states, “Using a machine learning approach”, is this ML approach different from that used in the prior sentences? The rest of this sentence seems to conflict with an earlier written sentence “our analyses revealed similarities in both diversity and composition between microbiomes of pts with CLL and pts of elective cardiac surgery. It seems the reference group is different depending on which group is used to compare to CLL. Why are you comparing to cardiac pts in the first place? They seem different than healthy controls

The abstract has been reformulated for better clarity and the involvement of samples from patients planned for elective cardiac surgery have been explained in the Importance and Methods section:

... Samples from patients with planned cardiac surgery were included as they represent severely diseased non-hematological cohort, only samples collected prior to the cardiac surgery were included due to possible changes in the microbiome composition upon surgery.

The sentence, “while we demonstrated that medication and underlying disease had a significant influence on ...”. The was not demonstrated in the abstract. The prior sentence, “when addressing microbiome variation, medication disease, etc”, isn’t clear and doesn’t support the sentence, “while we demonstrated that ...”.

Thank you for pointing this out, the abstract has been reformulated for improved clarity.

The text for Importance is written better but with more clarity still needed:

1. The sentence, “Pts receiving intensive cancer treatment had least diverse microbiomes”. Who are these Pts. CLL? Other cancers?
2. The sentence, “Using machine learning, this ...” didn’t you use machine learning to support the prior sentence?
3. “The findings contribute to microbial biology by expanding...”, Is this microbial biology of CLL pts or in general?
4. When using the word Patients, it’s not clear if referring to CLL patients or some other patients. Please be more specific.

The Importance section has been reformulated, we now refrain from using other hospital cohort, etc. and use more specific terms.

Cohort details are now also provided in the Importance section.

This study reveals how disease and medication influence the gut microbiome in patients with chronic lymphocytic leukemia (CLL) when compared to other more severe hematological malignancies, cohort of patients scheduled for elective cardiac surgery representing severely diseased non-hematological cohort, and a cohort of healthy individuals. ...

Please clarify when the fecal samples were collected in the CLL patients and other patients. In particular, were samples collected after respective cancer treatment.

Sample collection details for CLL were added to the new Supplemental Table 1 and mentioned in the Methods section:

...Samples in the CLL cohort were collected at different stages of the disease (including treatment naïve, treated and relapsed-refractory patients, Supplemental Table 1). ...

Feature		All (n=85)
Time from diagnosis to sampling	years (median, IQR)	1.9 (0.3, 5.3)
Binet stage at diagnosis	A	72 (85%)
	B	10 (12%)
	C	3 (3%)
Treatment status at sampling	Treatment naive	64 (75%)
	Treated	18 (21%)
	Relapse-refractory	3 (4%)
Treatment type*	Chemotherapy (FCR, RCHOP+HD-MTX)	12 (14%)
	Chemoimmunotherapy (R+Bendamustin)	4 (5%)
	Targeted therapy**	5 (7%)
Immunoglobulin treatment before microbiome sampling	Treated	7 (8%)
	Not treated	78 (92%)
IGHV	M-CLL	44 (52%)
	U-CLL	36 (42%)
	NA	5 (6%)
TP53	Mutated	10 (12%)
	Unmutated	65 (76%)

Pre-cardiac surgery patients:

... Samples from patients scheduled for elective cardiac surgery, included as a critically ill non-hematological contrast cohort to CLL, were collected before surgery to avoid potential microbiome changes caused by the procedure. ...

A citation has now been added to clarify the methods section including specifying the time point for collecting pre-HSCT samples:

Only samples collected at least 5 days prior to aHSCT (due to start of the pre-transplantation conditioning regimens) were included; detailed sample collection in the pre-aHSCT groups has been previously described by Ilett, et al (23).

Why was medication collected only between 30 days and 1 day prior to fecal sample date? Is there data to support that the microbiome recovers after 30 days of medication?

We thank the reviewer for raising this important point which was based on extensive discussions with the clinicians. We chose to collect medication data within the window of 30 days to 1 day prior to fecal sampling to capture exposures most likely to have a direct impact on the microbiome composition at the time of sampling. We recognize that the duration of microbiome recovery following medication is highly variable, and there is conflicting evidence on the time of recovery in the literature:

For example, Palleja et al. (DOI: 10.1038/s41564-018-0257-9) showed that after a course of broad-spectrum antibiotics, microbial diversity largely recovered within ~6 months, but some taxa never returned to baseline.

Similarly, Dethlefsen and Relman (<https://doi.org/10.1128/mbio.01693-15>) observed that certain community members recovered within weeks, while others remained perturbed for months, indicating incomplete or prolonged recovery.

In contrast, Hasegawa et al. (DOI: 10.1093/cid/civ137) showed that recovery can occur as early as two weeks in some contexts.

Nguyen et al. (DOI: 10.1016/j.celrep.2022.110649) provided evidence that recovery trajectories may extend up to one year, depending on the medication and host factors.

We also added a sentence to Discussion section to make reader aware of the long-term effect of antimicrobials usage:

... However, the near-perfect performance of the first classifier may also reflect the exclusion of the pre-cardiac surgery cohort, whose microbiome profiles closely resembled those of CLL patients, which could also be impacted by long term effect of antimicrobials more frequently used in the CLL cohort before and after diagnosis of CLL (47).

In addition, by restricting to this window, we also exclude individuals on long-term medications (e.g., proton-pump inhibitors, antidiabetic drugs), because if a person is on chronic medication, we assumed it would be recorded in the 30-day window.

104 individuals with pre aHSCT are MDS or “other hematological diseases”. What are the other hematological diseases? Myeloid? Lymphoid? How representative are these individuals with hematological malignancies?

Diagnoses with numbers of patients from the other pre-aHSCT cohort are now detailed in new Supplemental Table 1.

Overview of other pre-aHSCT cohort diagnoses	
Disease type	n
Myelodysplastic syndrome (MDS)	65
Severe aplastic anemia	5
Non-Hodgkin lymphoma (NHL)	9
Chronic myeloid leukemia (CML)	6
T-cell lymphomas	7
Childhood acute lymphoblastic leukemia	6
Other	6

Why are the 9 kidney donors a separate cohort? Why are they not combined with healthy controls?

We kept the kidney donors as a separate cohort because they were substantially younger than the healthy controls. Age is a known factor influencing microbiome composition, so combining these groups could confound the analysis. In contrast, the healthy control cohort was age-matched to the CLL patients (not perfectly, but this was the closest match we could achieve), allowing for more appropriate comparisons.

The basis for including patients pre cardiac surgery is not justified. Why does it matter that CLL patients have microbiome similar to pre-cardiac surgery patients. The meaning of this is unclear.

We included patients scheduled for elective cardiac surgery as a comparison cohort to represent critically ill individuals unrelated to hematological disease. This allowed us to test whether being critically ill in general could drive microbiome changes like those observed in CLL, which helped to separate the effects of hematological disease from illness-related microbiome perturbations. The comparison is not intended to imply direct similarity between CLL and cardiac patients, but rather to evaluate whether critical illness alone could account for observed microbial patterns.

Table 2 is not adding much value. It's showing raw counts of medication usage across medication types and cohorts. It should be a supplemental table. I would add a row indicating no medications. This is because it seems to be implied that everyone in the pre-cardiac or kidney donor cohorts had a least one medication (based on Figure 3). One cannot tell this information based on data in Table 2.

Table 2 was updated to include the number of patients in each cohort who did not receive any medication in the 30 days prior to sampling to avoid the false assumption that all individuals had taken medication.

As Table 2 is referred to multiple times throughout the manuscript, and we believe it is important to be transparent about which medication classes were inspected and used for exclusion criteria in the machine learning analyses of CLL microbiome patterns, as well as when assessing the effect of medication on the microbiome in the final part of the manuscript, we decided after thorough consideration to keep Table 2 as part of the main manuscript. The counts of patients using different medications also illustrate polypharmacy and support the discussed limitation regarding varying medication regimens across cohorts.

Figure 1.A: Please add in statistical tests comparing between the cohorts. I suggest using the healthy controls as the reference group.

Thank you for this suggestion, the Figure 3 was updated to include the statistical testing.

Figure 1B: interesting figure but not sure if these differences in angle are statistically different from CLL (0 degrees) given the high SD values.

We agree that the relatively high SD values need careful interpretation. As the reviewer suggests, significance can be evaluated in relation to whether the 95% CI (approximately ± 2 SD) overlaps with 0° . Based on this approach, the pre-cardiac surgery group shows only a trend toward significance since the CI overlaps with 0° , whereas the other groups demonstrate non-overlapping CIs with 0° , indicating a statistically significant deviation from the CLL reference. This pattern is consistent with our interpretation that the CLL and pre-cardiac surgery groups tend to have more similar microbiome compositions (also shown demonstrated in Fig 2A), whereas the other groups diverge significantly from CLL.

Figure 3: Why compare CLL to the combined non-cancer and healthy controls. The basis of this comparison is not justified. The second part of Figure 3 (CLL vs healthy controls) is not novel, per se, but rather an extension of the authors' prior work. Thus, the Discussion should be reduced or clarified as to what is gained by these additional analyses.

Thank you for pointing out the need to clarify this. A new Supplemental Figure 2 was added to show the sequence of events that led to the current version of Figure 3. The original aim was to validate and refine the CLL-specific microbiome pattern. With the availability of medication data, we further aimed to refine the microbiome pattern while accounting for the known influence of medications. Supplemental Figure 2A shows that the pre-aHSCT cohorts had to be excluded because chemotherapy heavily influenced the classification of CLL based on microbiome profiles. Supplemental Figure 2B supports our hypothesis that medication use can mask microbiome pattern identification; classification improved when we excluded cohorts with medication use in the 30 days prior to sampling.

We acknowledge that the second part of Figure 3 (CLL vs. healthy controls) extends our prior work, and we have clarified in the Discussion that the new analyses primarily aim to validate/refine previous findings while controlling for medication effects, rather than presenting entirely novel patterns.

Also it is unknown whether the signature is driven by the CLL disease or the signature drives the disease. Should be stated in limitations and need for future prospective studies.

Thank you for pointing this out, which we fully agree on, discussion section has been modified.

... Importantly, it remains unclear whether this microbiome signature is a consequence of CLL or whether it contributes to disease development, which should be acknowledged as a limitation. While our previous mouse model study provides initial evidence supporting a causal role of the microbiome in CLL (24), more prospective studies are required to clarify the causal direction of these associations.

Additional statistical test should be provided

Additional statistical testing has been added also based on Reviewer 1 suggestions.

Re: Spectrum00944-25R1 (Explainable Machine Learning to Identify Chronic Lymphocytic Leukemia and Medication Use Based on Gut Microbiome Data)

Dear Dr. Carsten Utoft Niemann:

Thank you for your attention to the reviewers comments.

Your manuscript has been accepted, and I am forwarding it to the ASM production staff for publication. Your paper will first be checked to make sure all elements meet the technical requirements. ASM staff will contact you if anything needs to be revised before copyediting and production can begin. Otherwise, you will be notified when your proofs are ready to be viewed.

Sincerely,
S. Wesley Long
Editor
Microbiology Spectrum